# PFTAIRE Kinase L63 Interactor 1A (Pif1A Protein) Is Required for Actin Cone Movement during Spermatid Individualization in *Drosophila melanogaster*

**DOI:** 10.3390/ijms23063011

**Published:** 2022-03-10

**Authors:** Harrison D. Pravder, Dorota Grabowska, Kaushik Roychoudhury, Betty Zhang, Deborah Frank, Przemysław Zakrzewski, Marta Lenartowska, Kathryn G. Miller

**Affiliations:** 1Department of Biology, Washington University in St. Louis, St. Louis, MO 63130, USA; harrison.pravder@yale.edu (H.D.P.); dorota.grabowska@wustl.edu (D.G.); roychoudhury@gmail.com (K.R.); bzhang199@gmail.com (B.Z.); miller@wustl.edu (K.G.M.); 2Department of Anesthesiology, Yale School of Medicine, New Haven, CT 06520, USA; 3Department of Radiology, Washington University School of Medicine in St. Louis, St. Louis, MO 63110, USA; 4Department of Obstetrics and Gynecology, Center for Reproductive Health Sciences, Washington University School of Medicine in St. Louis, St. Louis, MO 63110, USA; dfrank22@wustl.edu; 5Department of Cellular and Molecular Biology, Faculty of Biological and Veterinary Sciences, Nicolaus Copernicus University in Torun, 87-100 Torun, Poland; p.zakrzewski@bristol.ac.uk; 6School of Biochemistry, Faculty of Life Sciences, University of Bristol, Bristol BS8 1TD, UK

**Keywords:** actin-binding proteins, actin cones, *Drosophila melanogaster*, male sterility, *Pif1A* gene, spermatid individualization, Z3-5009 mutant

## Abstract

A useful model for determining the mechanisms by which actin and actin binding proteins control cellular architecture is the *Drosophila melanogaster* process of spermatogenesis. During the final step of spermatogenesis, 64 syncytial spermatids individualized as stable actin cones move synchronously down the axonemes and remodel the membranes. To identify new genes involved in spermatid individualization, we screened a collection of *Drosophila* male-sterile mutants and found that, in the line Z3-5009, actin cones formed near to the spermatid nuclei but failed to move, resulting in failed spermatid individualization. However, we show by phalloidin actin staining, electron microscopy and immunocytochemical localization of several actin binding proteins that the early cones had normal structure. We sequenced the genome of the Z3-5009 line and identified mutations in the PFTAIRE kinase L63 interactor 1A (*Pif1A*) gene. Quantitative real-time PCR showed that Pif1A transcript abundance was decreased in the mutant, and a transgene expressing Pif1A fused to green fluorescent protein (GFP) was able to fully rescue spermatid individualization and male fertility. Pif1A-GFP localized to the front of actin cones before initiation of movement. We propose that Pif1A plays a pivotal role in directing actin cone movement.

## 1. Introduction

Within multicellular organisms, cells adopt a wide variety of shapes to allow them to function properly. A key component of cellular architecture is the actin cytoskeleton, made up of filaments (F-actin) formed via polymerization of globular actin. F-actin assembly and dynamics are regulated by numerous well-conserved actin binding proteins. A useful model for determining the mechanisms by which actin and actin binding proteins control cellular architecture is the *Drosophila melanogaster* process of spermatogenesis. Mitotic divisions, incomplete cytokinesis and meiosis produce 64 interconnected spermatids [1,2] surrounded by two somatic cyst cells. The cysts then elongate as they elaborate axonemes [3,4,5]. Finally, the spermatids individualize to form individual haploid male gametes. At the start of individualization, dense F-actin forms around each spermatid nucleus. The actin then remodels to form an actin cone (also known as an investment cone), and the 64 actin cones travel synchronously down the axonemes. As the cones travel, the membrane is reorganized so as to encase each spermatid in its own membrane tightly juxtaposed to the axoneme, and the cytoplasmic components are pushed out ahead of the actin cones, forming a cystic bulge [4].

Actin cones have two domains. The rear domain contains long actin bundles that run parallel to the axoneme [6,7], whereas the front contains a dense actin meshwork. A number of well-studied actin-associated proteins localize specifically to these two structural domains. For example, the actin-bundling protein quail initially localizes throughout developing cones and relocates solely to the rear domain upon cone movement [7]. The Arp2/3 complex and its activator cortactin, which mediate actin filament branching, initially localize throughout the cones and then concentrate at the front as the cones begin moving [7,8]. Finally, the minus-end-directed actin motor protein myosin VI localizes to the front of cones and helps stabilize the cone shape and function [6,7,8,9,10].

One of the basic unanswered questions is how cone movement commences and persists. Actin-polymerization inhibiting drugs, but not microtubule-affecting drugs, disturb actin cone movement [4], suggesting that F-actin dynamics are necessary for cones to travel along the axonemes. Electron microscopy imaging shows that, before synchronous cone movement, only actin bundles are present around the spermatid nuclei [7]. Moreover, mutations that inhibit bundle formation allow cones to form but not to move, indicating that the rear domain is primarily responsible for cone movement. Conversely, if the front meshwork is not formed, cones consisting only of bundles can move, but the cones are not able to exclude the cytoplasm, and proper membrane remodeling does not occur [7]. These and other observations indicate that actin polymerization, turnover and stability must be properly regulated to permit actin cone formation and movement during spermatid individualization.

Here, we aimed to identify additional proteins involved in actin cone dynamics. We describe identification of a mutation that impairs actin cone movement and present evidence that the gene responsible is PFTAIRE kinase interactor 1A (Pif1A).

## 2. Results

### 2.1. Z3-5009 Mutants Have Defective Spermatid Individualization

To identify candidate genes required for actin organization and function during *Drosophila* spermatid individualization, we obtained extant male sterile lines from an ethyl methanesulfonate mutagenesis screen [11]. We performed staining to visualize actin in the testes of 600 homozygous mutants that had defects in post-meiotic stages and focused on those with actin cone defects including scattered actin cones, arrested cones, absent cones and cones that did not move away from the nuclei.

One line, Z3-5009, had a strong phenotype in which actin cones formed in association with spermatid nuclei in the basal portion of the cyst (Figure 1a, inset) and looked similar to those in testes dissected from wild type (WT) males (Oregon R, Figure 1b, inset 1). However, no actin cones progressed away from the nuclei (Figure 1a) as normally occurs in WT testes (Figure 1b, inset 2). Quantitative analysis revealed that the testes from Z3-5009 mutants had significantly more nuclear bundles without actin cones and more nuclear bundles associated with actin cones than testes from WT flies (Figure 1c). Conversely, testes from Z3-5009 males contained no cystic bulges/waste bags (Figure 1c), which can be seen in testes from WT flies (Figure 1b, inset 3). These results suggest that actin cone movement did not occur in the Z3-5009 mutant.

If actin cone movement failed in the Z3-5009 mutant, then we would expect that spermatid individualization would also fail. To assess this possibility, we performed electron microscopy on cross-sections of testes. In testes from WT flies, individualized spermatids can be seen in which each axoneme is juxtaposed to mitochondrion and surrounded by a membrane (Figure 2a). In contrast, cross-sections of testes from Z3-5009 mutants showed numerous axonemes and mitochondria within a single membrane, and the axonemes were not tightly juxtaposed to the mitochondria (Figure 2c), thus we conclude that spermatid individualization failed in Z3-5009.

### 2.2. Actin Cones Form Correctly but Do Not Move in Z3-5009

To identify actin defects that prevented actin cone movement in Z3-5009 males, we first imaged the actin cones near spermatid nuclei at higher magnification. These newly forming cones are usually located at the apical end of the testes (Figure 3a). In WT flies, the fronts of actin cones are only slightly larger than the rear when the cones are in proximity to the nuclei (Figure 3b, [6,7]). As the cones move away from the nuclei, they adopt an obvious cone shape in which the front becomes substantially wider than the rear (Figure 3c, [6,7]; Figure 3g vs. Figure 3f). In testes from Z3-5009 flies, some of the actin cones appeared to move slightly away from the spermatid nuclei, but were never observed more than 100 µm from the nuclei (Figure 3k,m’). These cones had wider fronts and narrower rears than actin cones near nuclei in WT testes.

Given that Z3-5009 actin cones in proximity to spermatid nuclei appeared to form correctly but were unable to move away from nuclei and migrate down the cyst, we wondered whether they had proper localization of two key actin binding proteins. First, we examined localization of myosin VI, which is required for spermatid individualization [6,12]. In WT testes, myosin VI localizes diffusely throughout actin cones before they begin to move (Figure 3d, [7]). Upon initiation of cone movement away from nuclei, myosin VI accumulates at the front edges of the cones (Figure 3d, [7]; Figure 3f, arrows) and remains there until the cones reach the waste bag [6,8]. In Z3-5009 mutants, myosin VI was similarly located at the front of actin cones that were near spermatid nuclei (Figure 3j, arrows).

Second, we examined localization of the actin-bundling protein quail, which is *Drosophila* ortholog of vertebrate villin [13]. In the early stages of cone movement in WT testes, quail localizes throughout the cones (Figure 3e, [7]). As cones move further from the nuclei, quail completely localizes to the rear of cones (Figure 3e, [7]; Figure 3g, arrows). In Z3-5009 testes, quail localized throughout the entire cones with higher density in the rear (Figure 3k, arrows). Together, these observations suggest that actin cones in Z3-5009 formed similarly to those in WT flies but were unable to move away from the spermatid nuclei.

### 2.3. Microtubule Dynamics and Caspase Activation Are Not Impaired in Z3-5009

Because we found no obvious actin binding protein defects in testes from Z3-5009 flies, we wondered whether other processes required for cone movement were defective. Thus, we examined axonemal and cytoplasmic microtubules. In WT testes, axonemal tubulin is modified by poly-glycylation before actin cone movement. To determine whether this occurred correctly, we stained testes from Z3-5009 flies with an antibody that recognizes poly-glycylated tubulin and found that the axonemes appeared similar in testes from WT and those from Z3-5009 flies (Figure 3h,l). In WT testes, cysts are full of cytoplasmic microtubules (which can be observed by staining with an anti-alpha tubulin antibody) before cone movement (Figure 3i), but these microtubules break down as the cones move away from the nuclei (Figure 3i’). In testes from Z3-5009 flies, we observed a similar effect; alpha tubulin staining was lower in cysts in which actin cones had moved slightly away from the nuclei than in cysts in which cones were close to the nuclei (Figure 3m,m’).

In WT testes, several caspases are activated upon actin cone movement, and caspase proteins accumulate in cystic bulges and waste bags, where they likely function to cleave proteins that accumulate ahead of the cones. Because actin cones form but do not move in some caspase mutants [14], we wondered whether Z3-5009 had defective caspase function. To test this idea, we stained testes with an antibody that recognizes cleaved (activated) caspase-3 [15] and found that testes from both WT and Z3-5009 flies had strong staining throughout the cysts (Figure 4). However, given that no cystic bulges or waste bags formed in testes from Z3-5009 males, we did not observe intensely stained cystic bulges or waste bags in the mutant (Figure 4b) as appear in WT cysts (Figure 4a). We conclude that the failure of cone movement in Z3-5009 mutant was not due to defective microtubule dynamics or caspase activation.

### 2.4. Actin Cone Ultrastructure Is Disrupted in Z3-5009 Testes

Given that we found no obvious defects in selected actin binding proteins, microtubule dynamics, or caspase activation in Z3-5009 mutant, we wondered whether actin cone ultrastructure was disrupted in these males. Actin ultrastructure in cones can be determined by isolating individualizing cysts, decorating actin filaments with rabbit skeletal myosin II subfragment 1 (S1) and performing electron microscopy. In WT flies, newly forming actin cones that are closely associated with spermatid nuclei are composed only of bundles (Figure 5a, [7]). When the cones start to move but are still near the nuclei, small regions of actin meshwork form at the fronts (Figure 5b, [7]). When cones have moved away from the nuclei, they contain two distinct structural domains: meshwork in the front and bundles in the rear (Figure 5c, [6,7,10]), and the front meshwork grows as the cones move during individualization (Figure 5d, [7,10]).

When we attempted to isolate individualizing cysts from Z3-5009 homozygous flies, the cysts appeared thin and wispy. Additionally, they were fragile and often ruptured during the S1 decoration procedure. Thus, we were only able to decorate and analyze a few early cysts from homozygous mutant males. In these cysts, we found only actin cones localized close to the spermatid nuclei (Figure 6a–d) and all of them were composed of much shorter and thinner actin filaments than the WT cones (Figure 6a–d vs. Figure 5a–d).

Some of the early Z3-5009 cones contained mainly parallel bundles of actin, with a higher density of actin at their fronts than in WT (Figure 6a vs. Figure 5a). Other mutant cones, which appeared to have moved slightly away from the spermatid nuclei, had a disturbed structure with a bigger front meshwork and a less organized rear region of parallel bundles than in WT early cones (Figure 6b,c vs. Figure 5b). Additionally, some of the early Z3-5009 cones had abnormal actin meshwork that was denser on one side of the cone than the other (Figure 6a,b, arrows). In other cases, the mutant actin cones were disorganized (Figure 6c,d), and we never observed moving cones with dense front actin meshwork in Z3-5009 homozygous cysts. Moving cones with dense front actin meshwork are typical for individualizing WT cones (Figure 5d). We conclude that actin cones in Z3-5009 form a rear bundle region and begin to form a front meshwork but are unable to form the dense meshwork associated with actin cone movement away from the spermatid nuclei.

### 2.5. Z3-5009 Phenotypes Are Caused by a Mutation in Pif1A

To identify the mutated gene(s) responsible for the Z3-5009 phenotypes, we isolated DNA from homozygous third instar larvae from the Z3-5009 strain as well as two other male-sterile mutants from the same collection, Z3-5662 and Z3-4257. We performed Illumina whole-genome sequencing, compared the genome sequences of the three lines, and identified exonic polymorphisms specific to Z3-5009. We then obtained commercially available deficiency lines that uncovered the locations of identified sequence polymorphisms and assessed their ability to complement the sterility phenotype. All of the deficiencies tested complemented Z3-5009 except for *Df(3R)ED5330*. This deficiency overlapped the region of four unique exonic point mutations in the genes *CG33191*, *Ir85A*, *Pyd* and *Rel*. We then obtained all of the commercially available lines containing deficiencies in this region. Two overlapping deficiencies, *Df(3R)Exe18143* and *Df(3R)Exe16150*, failed to complement Z3-5009 (Figure 7).

Three deficiencies failed to complement Z3-5009 overlap in the region of 8,669,632 to 8,834,050, encompassing cytological bands 85A5-85A11 and 85B1-85B2. Neighboring deficiency lines *Df(3R)BSC466*, *DF(3R)BSC197* and *Df(3R)BSC506* complemented Z3-5009, further narrowing the region of interest to 8,747,684 to 8,791,354, encompassing cytological bands 85A10, 85A11 and 85B1 (Figure 7, Table 1). This small region contains ten predicted protein-coding genes: *CG8223*, *CG34135*, *CG11768*, *CG8236*, *CG13318*, *CG34301*, *Pif1A*, *Pif1B*, *CG33191* and *CG33189*. Our initial sequencing analysis had revealed a single C>T base change in an exon of *CG33191*. We reexamined the Z3-5009 sequencing data to identify all polymorphisms in the region identified by our deficiency mapping and found a single C>T polymorphism directly downstream of the stop codon of Pif1A transcript RI (Figure 7 and Figure 8). No polymorphisms were identified in any of the other nine genes in the Z3-5009 genome, suggesting that the Z3-5009 phenotypes were due to mutations in *CG33191* or *Pif1A*.

To confirm whether mutations in *CG33191* or *Pif1A* caused the Z3-5009 phenotypes, we ordered RNA interference (RNAi) lines that knock down the expression of each of the genes in the region. We crossed them with flies expressing *bamGal4*, which drives expression in late spermatogonia and early spermatocytes [16]. We were unable to drive expression of CG33191-RNAi in males because both the *bamGal4* and the RNAi transgenes were located on the X-chromosome. BamGal4>Pif1A-RNAi and bamGal4>CG8236-RNAi males were mostly sterile, whereas all the other bamGal4>RNAi males were fertile (Table 2).

To determine whether mutations in *CG8236* or *Pif1A* might be responsible for the Z3-5009 phenotypes, we performed actin staining of testes from these flies. Testes from bamGal4>CG8236-RNAi had a scattered nuclei phenotype that did not mimic that seen in Z3-5009 mutants (data not shown). However, testes from bamGal4>Pif1A-RNAi flies had the same phenotype as the Z3-5009 mutant in most examined testes. Specifically, testes from these flies contained arrested actin cones associated with nuclei and F-actin sleeves (Figure 9a). In a few cases, we observed testes from bamGal4>Pif1ARNAi flies that looked similar to those in WT flies. However, this is consistent with the fact that 29% of the bamGal4>Pif1ARNAi males were fertile (Figure 9b). Together, this evidence suggested that a mutation in *Pif1A* was responsible for the Z3-5009 phenotype.

We hypothesized that the mutation of 45 nucleotides downstream of the stop codon of the Pif1A-RI transcript affected expression of this transcript in Z3-5009 flies. To test this idea, we performed quantitative real-time polymerase chain reaction (qPCR) experiments and found that testes from both Z3-5009 and bamGal4>Pif1A-RI-RNAi flies contained substantially less Pif1A-RI transcript than testes from WT flies (Figure 8c). Finally, to confirm that *Pif1A* was the gene responsible for the Z3-5009 phenotype, we generated a construct in which GFP was fused to the C-terminus of the Pif1A protein. This construct contained an HS83 promoter and the myosin VI 3′ UTR to drive constitutive, stable expression in the testis [12]. We examined the testes of flies that had this Pif1A-RI-GFP transgene and were homozygous for Z3-5009 (P[Pif1A-RI-GFP/CyO3; Z3-5009/Z3-5009]) and found that the Z3-5009 phenotypes were rescued, as demonstrated in four assays. First, these testes contained actin cones that migrated away from the nuclei (Figure 10a,f,g). Second, the testes had normal appearing actin cones as visualized by S1 decoration and electron microscopy (Figure 6e–h). Third, cross-sections of the testes showed tightly juxtaposed axoneme/mitochondria pairs surrounded by membrane (Figure 2b). Finally, homozygous Z3-5009 males that carried the transgene were fertile (Figure 9b).

### 2.6. Pif1A Associates with Nascent Actin Cones

To begin to define the role of Pif1A during *Drosophila* spermatid individualization, we examined localization of Pif1A-GFP in testes from P[Pif1A-RI-GFP]; Z3-5009/Z3-5009 flies. Pif1A-GFP localized to the fronts of actin cones as they were starting to form near nuclei (Figure 10c–e). However, Pif1A-GFP was absent from cones that had moved away from nuclei (Figure 10f,g). To confirm that the GFP-tagged Pif1A localized in a similar manner as the endogenous Pif1A, we obtained a line of flies expressing GFP within a *Minos* mediated integration cassette. This line (56701) has GFP inserted into the genome in-frame within exon 5 of *Pif1A*, which is expressed in four of the Pif1A transcripts, but not Pif1A-RI. In this line, GFP was also evident on early forming cones (Figure 10i–k) but absent from cones that had moved away from nuclei (Figure 10l,m). This finding suggests that multiple Pif1A isoforms are involved in actin cone movement.

## 3. Discussion

In this study, we have identified a new player, Pif1A protein, that may be important in actin organization, dynamics and function during the last step of *Drosophila* spermatogenesis. Specifically, we show that the male sterile line Z3-5009 contains mutations in the *Pif1A* gene and that the sterility and defects in spermatid individualization in this line can be rescued by expressing *Pif1A* in developing spermatids. We show that Pif1A protein localizes to the front of nascent, grows actin cones and disappears as the cones initiate movement away from the spermatid nuclei during individualization. We also show that in the Z3-5009 mutant, actin cones appear to form and mature almost normally but do not move along the sperm tails. This reveals that Pif1A affects an early step of actin cone movement.

Pif1A appears to affect actin cones in a manner that is distinct from other proteins that are required for actin cone movement during spermatid individualization. For example, flies carrying mutations affecting the actin polymerization protein profilin have an individualization defect in which actin cones form but do not move [7]. These mutant actin cones are structurally abnormal and lack rear actin bundles. Moreover, WT testes that were cultured in the presence of the caspase inhibitor Z-VAD displayed many early-stage actin cones in association with the spermatid nuclei, but showed no evidence of cone movement [14]. In contrast, Z3-5009 mutant cones appear to assemble normally, forming both a rear bundle domain and front meshwork that appeared normal at the ultrastructural level. Our previous work showed that actin bundles are important for cone movement, as cones lacking bundles do not move, whereas cones lacking meshwork can move [7]. Moreover, localization of several actin binding proteins, caspase activation and breakdown of cytoplasmic microtubules all occurred normally in Z3-5009 mutants. Together, these data suggest that actin bundles are necessary but not sufficient for actin cone movement along the axonemes and that Pif1A regulates or participates in the cone movement directly. Future work should assess the role of Pif1A in actin dynamics during spermatid individualization.

Our data are largely in agreement with those of Yuan et al. [17], who used CRISPR/Cas9 to create a frame shift nonsense mutation in the *Pif1A* gene. These authors report that *Pif1A* mutant flies develop actin cones, but the cones do not move, spermatid individualization fails and the flies are completely sterile. However, we note a few differences between our results and those of Yuan et al. [17]. Firstly, they occasionally observed actin cones facing the wrong direction, which we have never seen. Secondly, they report that tagged Pif1A protein localized to spermatid nuclei, whereas we observe GFP-tagged Pif1A at the fronts of actin cones. This difference may reflect the fact that we included a myosin VI 3′ UTR sequence on our construct, which we have found to increase transgene expression in late spermatids. We confirmed this localization by examining a line of flies in which GFP was inserted into the genome in-frame within exon 5 of *Pif1A*. In this line, GFP-tagged Pif1A also localized to the fronts of actin cones. Importantly, the GFP-tagged Pif1A transgene that we made also rescued the spermatogenesis defects. Therefore, it seems likely that endogenous Pif1A localizes to the fronts of actin cones. Future experiments, such as with an antibody specific to Pif1A, should be conducted to confirm this conclusion.

The exact function of Pif1A during spermatogenesis is unclear at this point. Yuan et al. [17] report that *Pif1A* is similar to the human gene coding CCDC157, which is associated with non-obstructive azoospermia. Additionally, they suggest that Pif1A protein is involved in regulating lipid metabolism genes, which could be important during spermatid membrane reorganization. To begin to address the function of Pif1A, we used the protein structure prediction program Phyre2 [18] and found that Pif1A was predicted to form two proximal domains connected to a central stalk-like alpha helical domain. This predicted structure was similar to that of proteins that are part of known actin- and microtubule-based motor complexes. Two of the top three hits, both predicted with 98.3% confidence, were the stalk region of the microtubule motor dynein and tropomyosin, an actin-binding regulator of the cytoskeleton. Moreover, yeast-two-hybrid screens have identified interactions of Pif1A with proteins involved in force-generating complexes [19], including dynactin 3, p24 subunit, a constituent of the dynein–dynactin complex. These structure prediction and interaction data are consistent with our findings suggesting that Pif1A regulates or participate in motor activity.

There are several ways in which a Pif1A motor complex may function. First, if the motor complex were microtubule-based, Pif1A could act as a regulator of or a linker between a microtubule-based motor (kinesin, dynein, etc.) that moves on the axoneme and the actin network of the cones. Pif1A may directly bind to actin or it may link to another as yet unidentified cone-associated protein. Therefore, loss of Pif1A would impair actin cone movement as the cones would not be able to connect to the force-generating protein. Second, Pif1A could act as a linker between the actin force-generating mechanism and a complex of other proteins that guides the cones along the axoneme. In this scenario, loss of Pif1A would impair actin cone movement as the cones would not be linked to the axoneme ‘track’. Both of these scenarios are consistent with our observation that both axonemal and cytoplasmic microtubules appeared normal in Z3-5009 mutants. Further study is needed to confirm or refute these possibilities.

## 4. Materials and Methods

### 4.1. DNA Sequencing and Analysis

Flies were raised on a standard agar cornmeal medium at either 25 °C or 18 °C. Homozygous third instar larvae were collected from Z3-5009, Z3-5662 and Z3-4257 stocks. DNA was isolated via standard protocols and processed at the Genome Technology Access Center (Washington University in St. Louis, MO, USA) for Illumina whole-genome sequencing. Mutations only found in Z3-5009 were compiled in a Microsoft Excel table. Silent mutations were eliminated from further analysis.

### 4.2. Transgene Construction

The coding region of transcript Pif1A-RI cDNA (GH05455) was amplified by PCR with forward primer 5′-ACTATACAAAGAATTCGCTGCCATGGGC-3′ and reverse primer 5′-GGTACTAGTGGATCCGTAGTATTTCTTAG-3′ (synthesized by IDT, Coralville, IA, USA). The Clone Tech InFusion System (Mountain View, CA, USA) was used to clone the PCR product into the CsprHS83 vector directly upstream and in frame with a sequence encoding GFP. Downstream of the inserted gene was a fly myosin VI 3′-UTR. Final vectors were confirmed by sequencing on premises (Biology Department, Washington University, St. Louis, MO, USA).

### 4.3. Fly Genetics

Genetic Services (Sudbury, MA, USA) generated transformant fly lines with the transgene P[Pif1A-RI-GFP] located at various loci on the first and second chromosomes. A second chromosome transformant with the most intense GFP signal was used to generate P[Pif1A-RI-GFP]/CyO; Z3-5009/Z3-5009 or P[Pif1A-RI-GFP]/P[Pif1A-RI-GFP]; Z3-5009/Z3-5009 flies.

*Minos* mediated integration cassette (MiMIC) line 56701 and all RNAi lines, including Pif1A-RNAi (KK106414), were obtained from the Bloomington *Drosophila* Stock Center (Indiana University, Bloomington, IN, USA). The *bam*-*Gal4* driver was a gift of Dr. Dennis M. McKearin [16].

### 4.4. Fertility Assays

Five Z3-5009 mutant virgin females were crossed with five males from each deficiency line in duplicate or triplicate. Three male Z3-5009/Deficiency progeny were then mated to three wild-type (WT, Oregon R) virgin females at 25 °C. After five days, the adult flies were removed. Fertility was assessed ten days later. Virgin *bamGal4* females (*bamGal4*; +; +) were crossed to males of the RNAi lines. Male progeny were tested for fertility as described above.

### 4.5. Immunofluorescence and Image Acquisition

Newly eclosed males (2–24 h after eclosion) were collected and paralyzed with carbon dioxide. Testes were dissected in phosphate-buffered saline (PBS) and fixed with 4% formaldehyde in PBST (0.1% Triton X-100 in PBS) for 15 min. Fixed testes were blocked in PBST supplemented with 3% BSA for at least 30 min. at room temperature or overnight at 4 °C. Then, the samples were incubated with primary antibodies rotating overnight at 4 °C. Primary antibodies included monoclonal mouse anti-*Drosophila* myosin VI antibody 3C7 (1:50, [20]), mouse anti-quail 6B9 (1:50, [7]), rabbit polyclonal anti-caspase-3 cleaved antibody (1:100, AB_2275226, PC679, EMD Biosciences, San Diego, CA, USA), mouse anti-α-tubulin (1:10, DM1A, Sigma-Aldrich, St. Louis, MO, USA), mouse anti-pan poly-glycylated tubulin (1:1000, AXO 49, MABS276, Millipore, Darmstadt, Germany) or polyclonal anti-GFP (1:300, AB_303395; ab290, Abcam, Cambridge, MA, USA). Testes were washed three times with PBST at room temperature, then incubated rotating for 1 h at room temperature with appropriate secondary antibodies and Alexa-568/488 conjugated phalloidin (1:100, Invitrogen, Carlsbad, CA, USA). Testes were washed three times in PBST and mounted on poly-L-lysine coated glass slides with Vectashield Mounting Media/DAPI.

Imaging was performed on a Nikon AI inverted confocal microscope with 405/488/561 nm lasers and Nikon NIS-Elements software. Images shown are single planes.

### 4.6. Scoring of Nuclear Bundle and Actin Cone Phenotypes

Imaging was conducted on a confocal Nikon A1 Inverted microscope. Each testis was scored for nuclei without associated cones (condensed nuclei that did not co-localize with actin cones), nuclei with cones (actin cones that were in register and colocalized with nuclear bundles), migrating cones (cone groups that were in register and did not colocalize with the spermatid nuclei bundle) and waste bags (rounded waste bags near the apical end of the testis).

### 4.7. Myosin Subfragment 1 (S1) Decoration and Electron Microscopy

Purification of rabbit skeletal myosin II and preparation of S1 subfragment followed conventional methods [21], and primary culture of dissected cysts was described previously [4]. For myosin II S1 fragment decoration, individualizing cysts were permeabilized with 0.1% saponin and treated with 2–4 mg/mL S1 fragment as described previously in [6]. S1-decorated cysts were fixed with 1% glutaraldehyde (Merck, Darmstadt, Germany) and 0.2% tannic acid (Electron Microscopy Sciences, Hatfield, PA, USA), post-fixed with 2% OsO_4_ (Merck, Darmstadt, Germany) and embedded in Poly/Bed 812 resin (Polysciences, Warrington, PA, USA). Ultrathin longitudinal sections were collected on copper grids, stained and examined by electron microscopy as described previously [10]. Cysts were examined at the following stages of spermatid individualization: very early cysts with actin bundles forming near the nuclei, early cysts with cones that were just initiating movement and individualized cysts with early or late cones that had moved away from the nuclei.

For cross-sections of spermatogenic cysts, testes were dissected from adult male flies and prepared for electron microscopy as described previously [6]. In brief, testes were fixed with 1.5% glutaraldehyde in 0.1 M PBS, pH 7.0, for 2 h on ice, postfixed in 1% OsO_4_ for 2 h at 4 °C and embedded in Poly/Bed 812 resin (Polysciences, Warrington, PA, USA). Ultrathin cross-sections were collected on copper grids, stained and examined on a Hitachi JEOL EM 1010 transmission electron microscope.

### 4.8. RNA Extraction and Quantitative RT-PCR (qPCR)

Total RNA was extracted from testes dissected from one-day-old males with TRIzol reagent (Sigma-Aldrich, St. Louis, MO, USA) and a Direct-zol RNA Miniprep kit (Zymo Research, Irvine, CA, USA) with DNase. cDNA was synthesized with random hexamer oligonucleotides and SuperScript II reverse transcriptase (Invitrogen, Grand Island, NY, USA). Quantitative real-time polymerase chain reaction (qPCR) assays were conducted with iTaq Universal SYBR Green Supermix (Bio-Rad Laboratories, Hercules, CA, USA) on a CFX Connect Real-Time PCR Detection System machine (Bio-Rad Laboratories, Hercules, CA, USA) according to the manufacturer’s suggested parameters. Measurements were made on three biological replicates for Oregon R, Z3-5009 and bamGal4 > Pif1ARNAi and one replicate for Pif1A-GFP; Z3-5009 and technical triplicates for each sample. The housekeeping gene ribosomal protein 49 (RpL32) was used as a control to normalize mRNA expression. Data were transformed and analyzed according to the ΔΔCt method with Bio-Rad CFX Manager software 3.0 (Bio-Rad Laboratories, Hercules, CA, USA). The following primers were used for qPCR: Pif1A-RI-forward: 5′-ATGAGGTACCGAGCCTCCA-3′; Pif1A-RI-reverse: 5′-GGGCTCTTTGCTTTGGAGA-3′; rp49-forward: 5′-CACCAAGCACTTCATCCG-3′; rp49-reverse: 5′-TCGATCCGTAACCG ATGT-3′.

## 5. Conclusions

Here we show that Pif1A protein is important in actin organization and function during the last step of *Drosophila* spermatogenesis. The Z3-5009 sterile males with mutations in the *Pif1A* gene show some defects in spermatid individualization which can be rescued by expressing *Pif1A* in developing spermatids. In WT males, Pif1A protein localizes to the front of actin cones and disappears as the cones initiate movement away from the spermatid nuclei during individualization. In contrast, the Z3-5009 actin cones appear to form and mature almost normally but do not move along the sperm tails. This reveals that Pif1A affects an early step of spermatid individualization, thus we propose that this protein plays a pivotal role in directing actin cone movement.

## Figures and Tables

**Figure 1 ijms-23-03011-f001:**
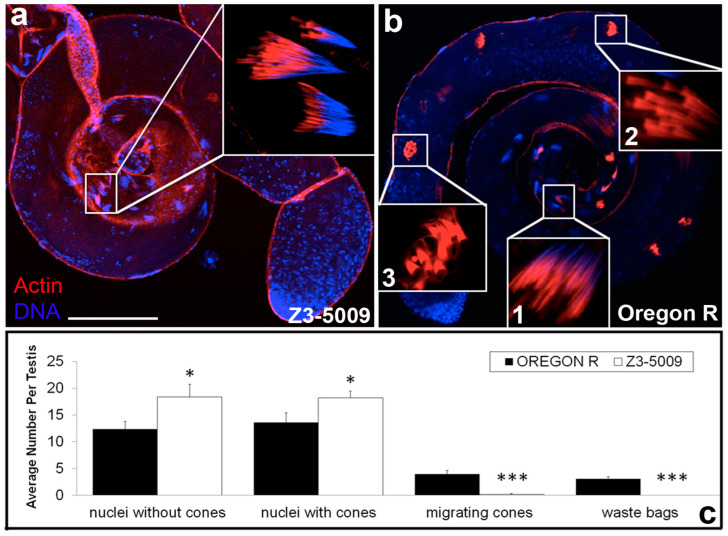
Spermatid individualization is disrupted in Z3-5009 mutant. Actin cone complexes in Z3-5009 testes (**a**) form in association with spermatid nuclei but do not progress past this stage (inset). In Oregon R control testes (**b**), actin cones form in association with sperm nuclei (inset 1), travel synchronously down the axonemes driving progress of spermatid individualization (inset 2, 3), and break down in a cystic bulge; phalloidin staining, red; DAPI staining, blue. (**c**) Classification of actin cone phenotype in control Oregon R (*n* = 13, black) and Z3-5009 mutant (*n* = 16, white); the average number of actin cones falling into each phenotypic class is reported. Bar 100 µm. * *p* < 0.05; *** *p* < 0.001.

**Figure 2 ijms-23-03011-f002:**
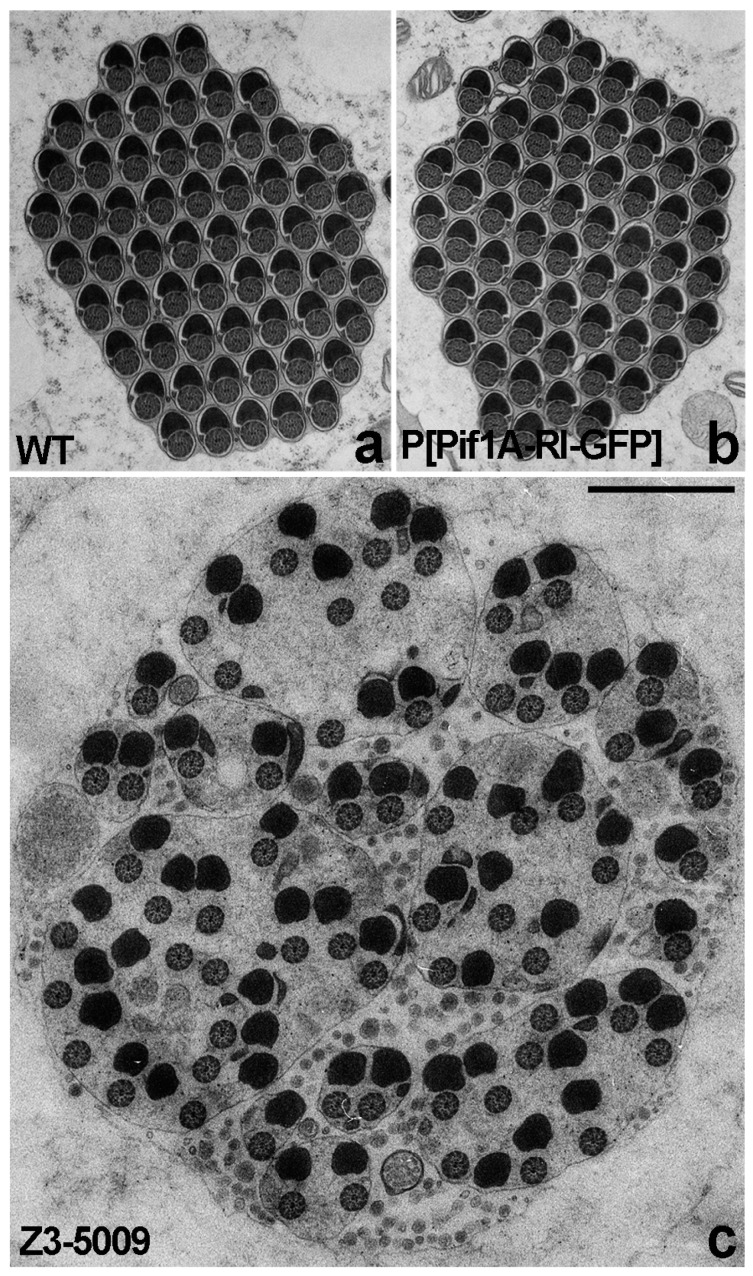
Individualization defects of Z3-5009 mutant visualized by electron microscopy. Cross-sectioned cysts of WT testis show individualized cysts (**a**); each of the control cysts contains pairs of axonemes and mitochondria derivatives enclosed by individual plasma membrane. Cross-sectioned cysts from Z3-5009 mutant show that individualization does not occur (**c**); numerous axonemes and mitochondria are enclosed within single membrane and surrounded by a large amount of cytoplasm, and the axonemes are not tightly juxtaposed to the mitochondria. Cross-sectioned cysts of flies that express the Pif1A-RI-GFP transgene and are homozygous for Z3-5009 (P[Pif1A-RI-GFP/CyO3; Z3-5009/Z3-5009]) show that the Z3-5009 phenotype is rescued—after individualization tightly juxtaposed axoneme/mitochondria pairs are surrounded by separate membranes (**b**). Bar 1 µm.

**Figure 3 ijms-23-03011-f003:**
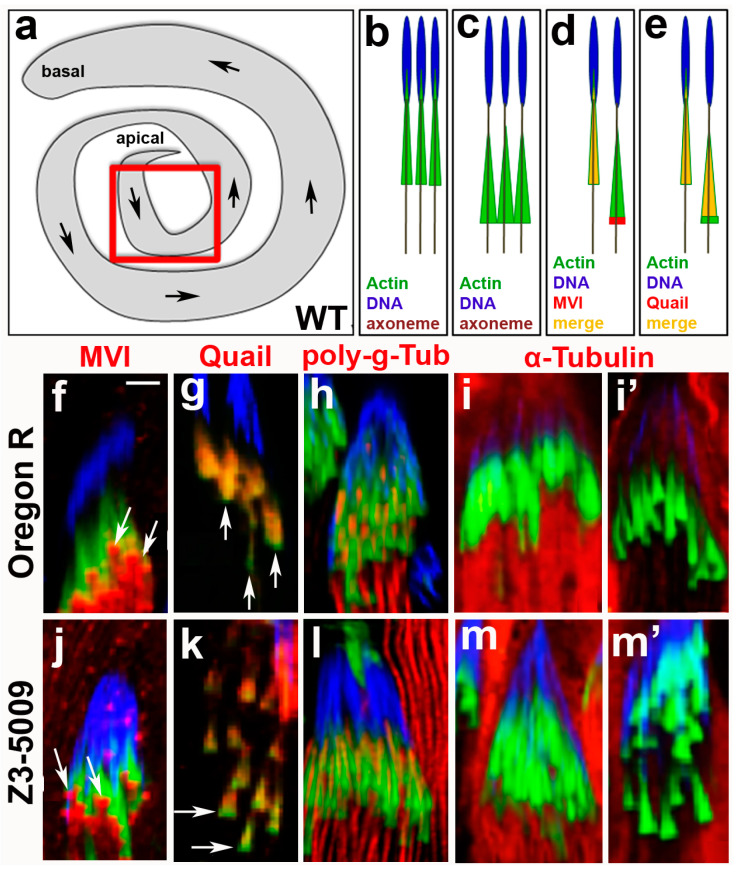
Immunocytochemical characterization of Z3-5009 mutant. Schematic representations of: whole *Drosophila* testis (**a**), newly forming actin cones in association with spermatid nuclei (**b**), early actin cones that have just moved a little away from the nuclei (**c**), myosin VI localization in newly forming/early control (WT Oregon R) cones (**d**), and quail localization in newly forming/early control cones (**e**); red box in (**a**) shows area-of-interest for all of the images in this figure, arrows in (**a**) indicate the direction of actin cones movement during spermatid individualization. Immunofluorescence comparative analysis of localization of selected proteins in control (**f**–**i’**) and Z3-5009 (**j**–**m’**) actin cones; phalloidin staining, green; antibody of interest, red; merge, yellow; DAPI staining, blue. Myosin VI localizes to the front of conical actin cones in Z3-5009 mutant ((**j**), arrows), consistent with the same-stage Oregon R cones ((**f**), arrows). Actin-binding quail appears to proceed towards the rear domains of the control (**g**) and Z3-5009 (**k**) cones as seen by *red* staining, leaving more intense actin (green) areas in the front of the cones (arrows in (**g**,**k**)). Spermatid axonemes are highlighted by poly-glycylated tubulin (**h**,**l**) and axoneme organization is normal in the Z3-5009 mutant (**l**) if compared to Oregon R (**h**). Alpha-tubulin appears to degrade to a similar extent in the same-stage Oregon R ((**i’**) vs. (**i**)) and Z3-5009 ((**m’**) vs. (**m**)) actin cones. Bar 5 µm.

**Figure 4 ijms-23-03011-f004:**
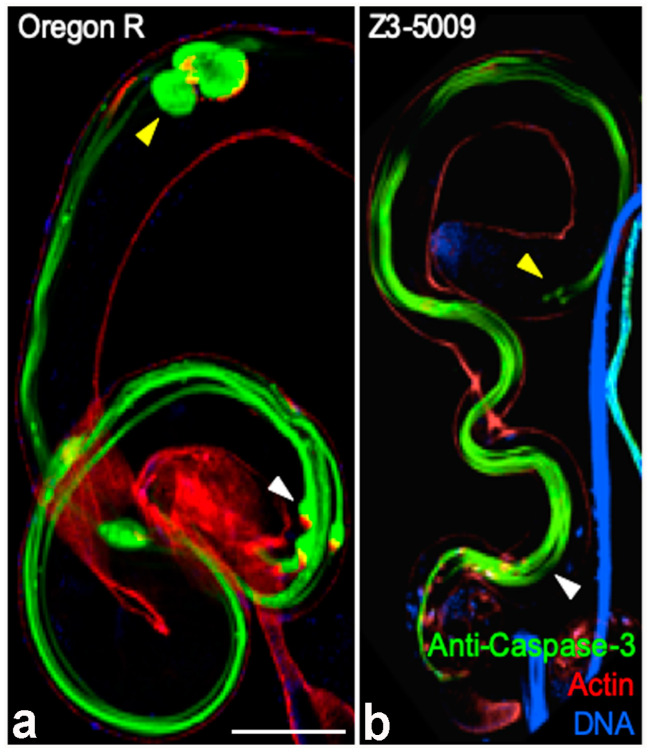
Activated caspase activity is normal in the Z3-5009 mutant. Activated caspases (green) are present throughout cysts in the Z3-5009 mutant (**b**), consistent with Oregon R (**a**). In control cysts, caspases accumulate in the cystic bulges ((**a**), white arrowhead) and waste bags ((**a**), yellow arrowhead). No cystic bulges ((**b**), white arrowhead) or waste bags ((**b**), yellow arrowhead) are seen in the Z3-5009 mutant, consistent with the observation that actin cones do not progress away from nuclei. Phalloidin staining, red; DAPI staining, blue. Bar 100 µm.

**Figure 5 ijms-23-03011-f005:**
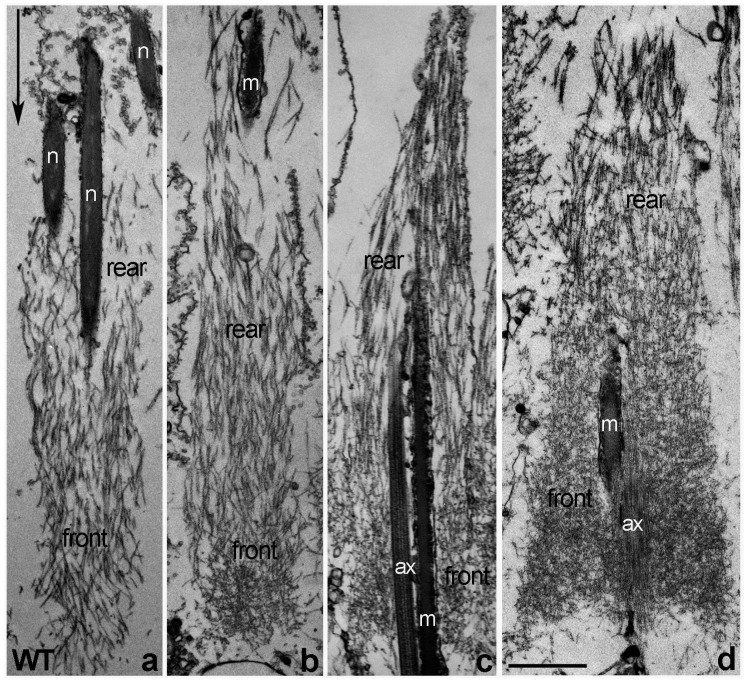
Ultrastructure of S1-decorated WT actin cones—longitudinal sections. Newly forming actin cones (in association with spermatid nuclei) are composed of actin bundles parallel to the longitudinal axis of the cone (**a**). A small actin meshwork is visible at the front of the early cones that have just moved a little away from the nuclei (**b**). Individualized actin cones have two structural domains (**c**,**d**): the rear domain is composed of actin bundles and the front domain is composed of a dense actin meshwork; as the cones move farther away from spermatid nuclei, the amount and extent of meshwork increases ((**d**) vs. (**c**)). Arrow in (**a**) indicates the direction of actin cones movement during individualization. Bar 1 µm.

**Figure 6 ijms-23-03011-f006:**
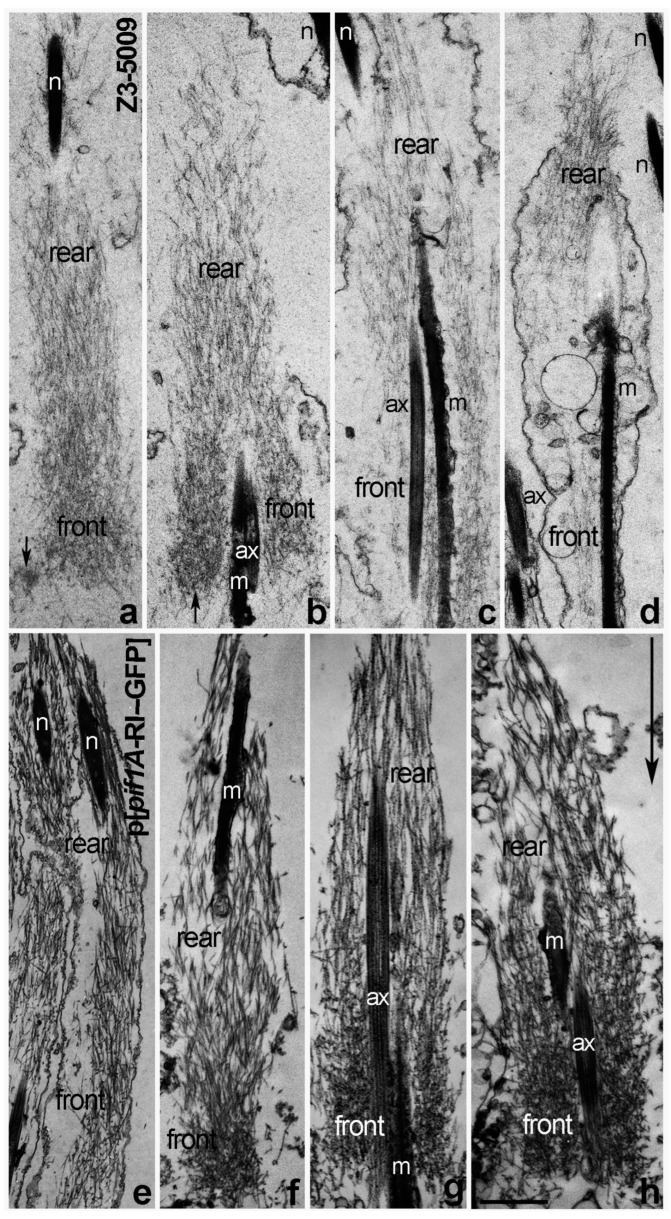
Longitudinal sections of S1-decorated Z3-5009 actin cones vs. cones from flies that express the Pif1A-RI-GFP transgene. Mutant actin cones (**a**–**d**) are composed of much shorter and thinner actin filaments than the WT cones (Figure 6a–d vs. Figure 5a–d). Newly forming Z3-5009 cones contained mainly parallel bundles of actin (**a**). Some of the mutant cones, which appeared to move slightly away from the spermatid nuclei (**b**), have disturbed structure with a bigger front meshwork and less organized rear region of parallel bundles than in WT early cones (Figure 6b vs. Figure 5b,c). Arrows in (**a**,**b**) indicate an abnormal actin meshwork that is denser on only one side of the early Z3-5009 cones. Most of the later mutant actin cones are disorganized (**c**,**d**). Longitudinal sections of actin cones from flies that express the Pif1A-RI-GFP transgene and are homozygous for Z3-5009 (p[Pif1A-RI-GFP/CyO3; Z3-5009/Z3-5009]) show that the Z3-5009 phenotype is rescued—the cones form and assemble actin properly during spermatid individualization (**e**–**h**). Arrow in h indicates the direction of actin cone movement during individualization. Bar 1 µm.

**Figure 7 ijms-23-03011-f007:**
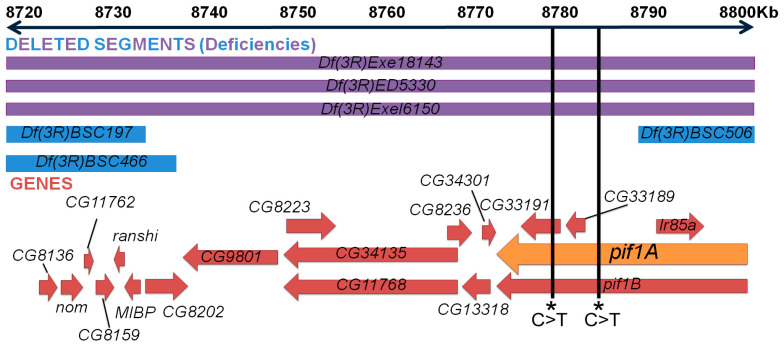
Identifying candidate genes for Z3-5009 phenotype. Cytological map of area-of-interest (chromosome 3R) derived by deficiency crosses. Top blue and purple rectangles refer to genetic regions that the deficiency lines are lacking—the sequences deleted extend beyond the constraints of this figure. Purple represents deficiency lines that failed to complement the mutant phenotype; blue represents lines that complemented (are fertile). In the lower portion, red arrows represent genes in the area-of-interest, and the orange arrow represents the gene thought to be responsible, *Pif1A*.

**Figure 8 ijms-23-03011-f008:**
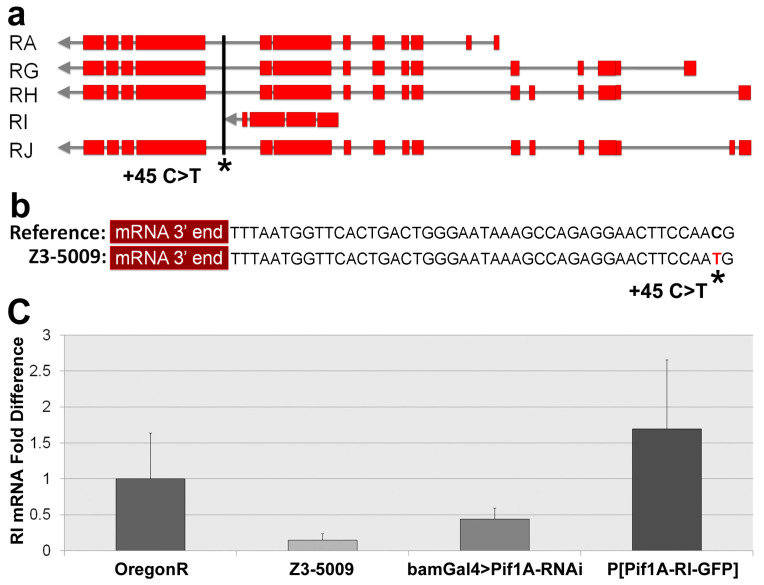
*Pif1A* overview and qPCR. Transcript map of the five predicted Pif1A transcripts (**a**). Exons of each transcript are shown in red; Pif1A-RI is the transcript used for rescue and targeting in RNAi. The SNP thought to be responsible for the phenotype is 45 bp downstream from the last codon of the Pif1A-RI mRNA transcript (**b**); it is a C>T transition. Relative Pif1A-RI mRNA expression levels in male testes (**c**); expression is decreased in Z3-5009 mutant. Error bars represent ± standard error.

**Figure 9 ijms-23-03011-f009:**
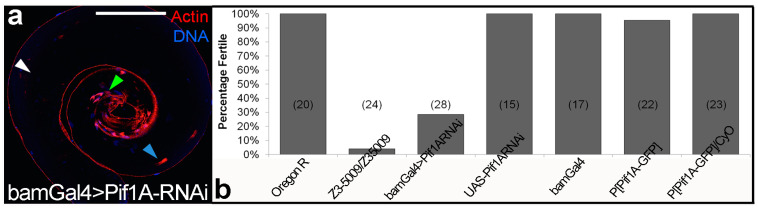
Spermatid individualization in bamGal4>Pif1A-RNAi testis and fertility assays. Actin cones are arrested with nuclear bundles in bamGal4>Pif1A-RNAi testis (**a**), consistent with the Z3-5009 mutant. Specifically, testes from these flies contained arrested actin cones associated with spermatid nuclei ((**a**), green arrowhead) or F-actin sleeves ((**a**), blue arrowhead) and no cystic bulges/waste bags are observed ((**a**), white arrowhead). Phalloidin staining, red; DAPI staining, blue. A small minority of the testes of this genotype appear roughly WT as explained in the text. Fertility assays (**b**)—all of the listed *Drosophila* lines represent males that were crossed with female virgin Oregon R flies; P[Pif1A-GFP], homozygote P[Pif1A-RI-GFP]. Numbers in parentheses refer to sample size. Error bars represent ± standard error. Bar 100 µm.

**Figure 10 ijms-23-03011-f010:**
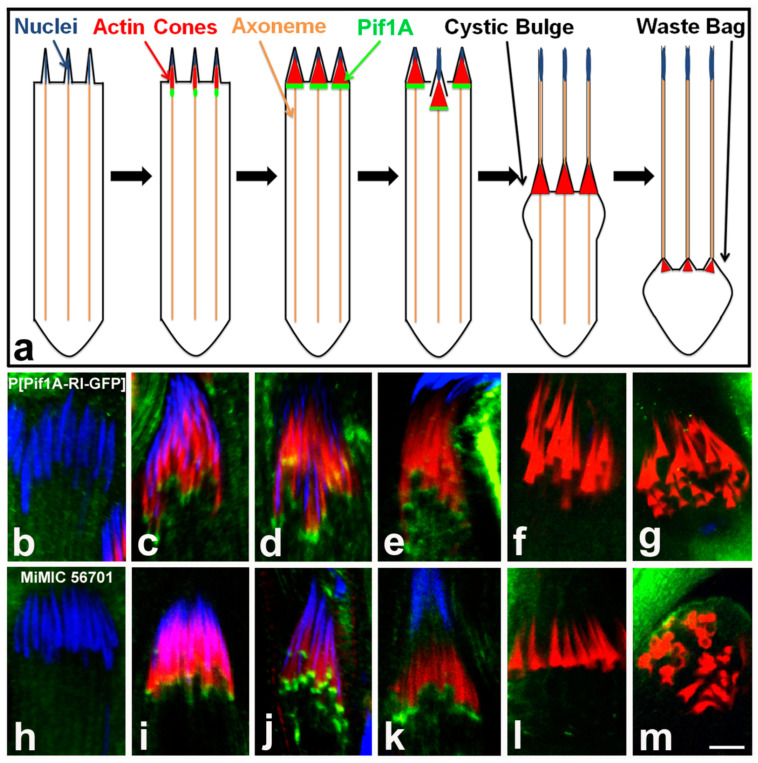
Pif1A localization time series. Schematic of localization of Pif1A protein throughout spermatid individualization (**a**); Pif1A (green) is present in the front region of nascent needle-shaped actin cones and is not observed in individualized cones. Representative homozygote P[Pif1A-RI-GFP] individualization complexes (**b**–**g**) stained with DAPI (blue), phalloidin/actin (red) and Pif1A-GFP (green) and representative MiMIC-GFP cones (**h**–**m**) stained the same way—localization profile is consistent with homozygote Pif1A-GFP transgenic cones. Bar 5 µm.

**Table 1 ijms-23-03011-t001:** Male fertility status of selected *Drosophila* deficiency lines.

Deficinecy Line	Fertility × OregonR
Df(3RBSC466	Fertile
Df(3R)BSC506	Fertile
Df(3R)BSC197	Fertile
Df(3R)Exe18143	Sterile
Df(3R)ED5330	Sterile
Df(3R)Exel6150	Sterile

**Table 2 ijms-23-03011-t002:** Male fertility status of selected *Drosophila* RNAi lines.

RNAi Line	Gene of Interest	BamGal4>GOI-RNAi × OregonR
VDRC 51028	*CG33720*	Fertile
P(KK110403)VIE-260B	*CG13318*	Fertile
W^1118-^, P(GD13849)v35860/TM3	*CG8223*	Fertile
P(KK114195)VIE-260B	*CG34135*	Fertile
P(KK114379)VIE-260B	*CG34301*	Fertile
P(KK110554)VIE-260B	*Pif1A*	Sterile
VCRD 51027	*Pif1B*	Fertile
P(KK112123)VIE-260B	*CG11768*	Fertile
P(KK107176)VIE-260B	*CG8236*	Sterile
P(KK106131)VIE-260B	*CG33189*	Fertile

## Data Availability

The data that support the present results are available from the corresponding author and Kathryn G. Miller upon reasonable request.

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
