# Peer review of "PFTAIRE Kinase L63 Interactor 1A (Pif1A Protein) Is Required for Actin Cone Movement during Spermatid Individualization in Drosophila melanogaster"

_ijms, 2022, doi:10.3390/ijms23063011_

Round 1

Reviewer 1 Report

Pravder HD and colleagues presented an interesting research paper aimed at identifying proteins involved in the organization and dynamics of actin during the spermatogenesis process in Drosophila melanogaster. The Authors have identified PFTAIRE kinase L63 interactor 1a (Pif1A) as required for actin cone moving during spermatid individualization. Though the manuscript is a little bit overloaded with analysis and figures, I am sure this area of research can benefit from this data.

I have only few minor comments that require clarification:

  1. Please check ʺZ3-5009ʺ throughout the text, e.g. in line 105: Z305009, in line 115: Z3-009.
  2. What is the difference between panels (a) and (b) in Figure 2? Should not panel ʺbʺ be a separate Figure in Section 5?

Could you mark axonemes and mitochondria derivatives?

  1. Line 140 – should be: (e) panel, not (d)
  2. In Figure’s caption 5 should be WT not Z30009. Am I right?
  3. Why cysts were decorated with conventional muscle myosin II subfragment not by myosin VI?
  4. What could have been the cause of wispy and fragile of cysts isolated from Z3-5009 homozygous flies?

Author Response

Dear Reviewer,

We would like to thank you for critically reading of our manuscript. We include below our point-by-point detailed responses to the reviewers’ comments. We believe the changes that we made in response to reviewers’ comments have improved the paper to be acceptable for publication in IJMS.

Reviewer 1:

Pravder HD and colleagues presented an interesting research paper aimed at identifying proteins involved in the organization and dynamics of actin during the spermatogenesis process in Drosophila melanogaster. The Authors have identified PFTAIRE kinase L63 interactor 1a (Pif1A) as required for actin cone moving during spermatid individualization. Though the manuscript is a little bit overloaded with analysis and figures, I am sure this area of research can benefit from this data.

I have only few minor comments that require clarification:

  1. Please check ʺZ3-5009ʺ throughout the text, e.g. in line 105: Z305009, in line 115: Z3-009.

We have done it. 

  1. What is the difference between panels (a) and (b) in Figure 2? Should not panel ʺbʺ be a separate Figure in Section 5?

Panel b shows the individualization phenotype of Z3-5009 homozygous mutant males that express the Pif1A-R1-GFP transgene in the testis. This transgene completely rescues the individualization defect. Although the rescue is described later in the results and the other panels are described here, we prefer to leave the ‘rescue’ panel in this figure to assist readers in interpretation: they need to compare the rescued cysts to both the wild-type and the mutant phenotypes. This single figure shows all those results.

  1. Could you mark axonemes and mitochondria derivatives?

We think that the axoneme's structure is well visible and if we write in the figure legend: "pairs of pairs of axonemes and mitochondria" it should be clear that the dark structure close to axoneme is mitochondrium. We prefer to do not mark these stryuctures to leave the photos so pretty as they are. 

  1. Line 140 – should be: (e) panel, not (d).

We have corrected this.

  1. In Figure’s caption 5 should be WT not Z30009. Am I right?

Yes, we have corrected this.

  1. Why cysts were decorated with conventional muscle myosin II subfragment not by myosin VI?

Mysoin II subfragment decoration is a well-known standard method to examine the organzation of actin in cells.  The technique reveals filament orientation and helps visualize structure more clearly than can be seen in undecorated actin filaments.  Mysoin VI decoration  has never been done or established as a technique for examination of actin strucutre and would not be appropriate for the purpose of better visualizing these structures.

  1. What could have been the cause of wispy and fragile of cysts isolated from Z3-5009 homozygous flies?

We suspect that there are actin defects caused by lack of Pif1A that affect the cells more broadly. This may cause an overall lack of stability of cells.

Reviewer 2 Report

In this manuscript, Pravder and colleagues describe the identification and characterisation of the Pif1A protein in spermatogenesis in Drosophila. The study is well-designed and pleasant to read.

Some comments remain.

Major comments:

  • The authors used only one RNAi line to target Pif1A. If possible, it would be beneficial to use a second one.
  • Is autophagy somehow impaired in Z3-5009 flies?
  • Line 255: why were those 2 specific mutant lines used as controls? Any specific reason?
  • Are all the Pif1A transcripts normally expressed in males? Is the expression of other transcripts affected in the Z3-5009 lines? Can the re-expression of transcripts other than RI also rescue the phenotypes?

Minor comments:

  • Line 105: Z305009 -> Z3-5009?
  • Are all the lines backcrossed to OreR?

Author Response

Dear Reviewer,

We would like to thank you for critically reading of our manuscript. We include below our point-by-point detailed responses to the reviewers’ comments. We believe the changes that we made in response to reviewers’ comments have improved the paper to be acceptable for publication in IJMS.

Reviewer 2:

In this manuscript, Pravder and colleagues describe the identification and characterisation of the Pif1A protein in spermatogenesis in Drosophila. The study is well-designed and pleasant to read.

Some comments remain.

Major comments:

  • The authors used only one RNAi line to target Pif1A. If possible, it would be beneficial to use a second one.

It is not possible to show another RNAi line, due to the combination of chromosmes needed to express these RNAi lines in individualizing cysts from the mutant. There is only one combination of appropriate driver and exisitng RNAi line that is available that can be expressed in the mutant animals. 

  • Is autophagy somehow impaired in Z3-5009 flies?

We did not test for autophagy defects.  However, caspase activity and other parts of the cell death machinery are important for individualization to proceed. Thus, cytoplasmic degradation may be important. Caspase activity was used to determine if that pathway is activated in the mutant.  It is normally activated in the Z3-5009 mutant, so at least this part of the ‘cellular degradation’ proces, which may be related to autophagy, is normal.  We have no information beyond this. 

  • Line 255: why were those 2 specific mutant lines used as controls? Any specific reason?

These are mutant lines that have the same genetic background and were isolated aspart of the same mutant screen.  These two lines have different defects in individualization as compared to Z3-5009 and they were being sequenced as part of a larger study of additional defects.  Any sequences in common among these lines are considered ‘wild-type.’ Sequences that differ in Z3-5009 from the other two are candidates to be changes that cause the Z3- 5009 defect.  Using this comparison, we could examine differences in sequence among them to identify potential unique changes in each line. 

  • Are all the Pif1A transcripts normally expressed in males? Is the expression of other transcripts affected in the Z3-5009 lines? Can the re-expression of transcripts other than RI also rescue the phenotypes?

We did not test any other transcripts for rescue or examine their expression. Expression of the R1 transcript is capable of complete rescue of the mutant phenotype. This is also the only transcript targeted by the RNAi experiment. The other transcripts would not be depleted by the R1 RNAi transgene expression. Males in which only the R1 transcript is knocked down have the Z3-5009 phenotype. Whether other transcripts are or are not expressed is irrelevant to our analysis, based on these results. As reviewer 1 points out, our paper already has a lot of data and analyses and adding more results that really are not relevant to our conclusions would not improve the paper or add more information of relevance.  

Minor comments:

  • Line 105: Z305009 -> Z3-5009?

This typo has been corrected. 

  • Are all the lines backcrossed to OreR?

Lines were not back crossed to OreR. Maintaining the mutant strains necessitates using balancer chromosomes that prevent recombination. In all the fertility experiments, mutant/transgene carrying males were crossed to OreR females.

Round 2

Reviewer 2 Report

In their revised manuscript and associated letter, the authors have answered all my previous concerns and questions. I don't have further comments on this manuscript that is informative on the topic, well written and pleasant to read. Looking forward to reading further studies on PifA1 and its role in spermatogenesis.